# Mental well-being in Swedish adolescents 2014–2023: A repeated population-based cross-sectional study focusing on temporal variations and differences between groups

Lena Uvhagen[1]*, Johanna Gustafsson[2], Fredrik Söderqvist[1]

**1** University Health Care Research Centre, Faculty of Medicine and Health, Örebro University, Örebro, Sweden, **2** School of Health Sciences, Faculty of Medicine and Health, Örebro University, Örebro, Sweden

* lena.uvhagen@oru.se

## Abstract

Mental well-being is more than merely the absence of mental illness; it is a multidimensional concept that includes both emotional and functional well-being, which are valuable resources during adolescence. In order to develop relevant interventions and policies to strengthen adolescent mental health, a continuous monitoring of the population well-being becomes important. The aim of the study was to examine the level, distribution, and changes in mental well-being over time in a Swedish adolescent population. Current study is based on four waves (2014-2017-2020-2023) of a cross-sectional student survey (N = 16288, $M_{age}$ = 16.23). The outcome was measured with the Mental Health Continuum Short Form. Ten explanatory factors were chosen to examine differences in mental well-being in the study population: Grade, Sex, Sexual orientation, Socioeconomic status, Country of birth, Visual, Hearing or Mobility impairment, Specific learning disorder and Neurodevelopmental disorder. Differences in mental well-being between groups as well as temporal trends were examined and evaluated through statistical testing and hierarchical multiple linear regressions modeling. Girls, non-heterosexual adolescents, and adolescents with low socioeconomic status or impairments have lower levels of mental well-being than boys, heterosexual adolescents, and adolescents with higher socioeconomic status or without impairments, respectively. A deterioration in mental well-being is seen over time for several groups; however, results of the multivariable analysis indicates that the deterioration is mainly an effect of sex and the significant decline in mental well-being seen among girls. The most significant factor for explaining the variation in mental well-being in this study is socioeconomic status. This study elucidates temporal changes and differences in levels of mental well-being between social groups in the adolescent population. The overall differences are small, but their potential implications for public health warrant careful consideration since they concern a significant part of the population. The results underscore the imperative of promoting mental well-being in adolescents, particularly among vulnerable groups.

**Data availability statement:** The data used in this study can be made available upon reasonable request for researchers who meet the criteria for access to confidential data. Data requests can be made to corresponding author or non-author contact person Professor Cecilia Åslund (cecilia.aslund@regionvastmanland.se). At this point, data cannot be shared publicly because of ethical and legal restrictions concerning the student's consent and data ownership. The data is owned by a third party, the County of Västmanland in Sweden, which have imposed restrictions.

**Funding:** The Public Health Agency of Sweden funded part of this work.

**Competing interests:** The authors have declared that no competing interests exist.

## Introduction

Mental well-being is a fundamental resource for feeling good, mastering everyday life, and enabling individuals' potential for productivity and personal growth [1]. This resource is especially important during adolescence, a period of biological, psychological, and social development. Adolescence is also a period of both positive mental health development and vulnerability to mental health problems [2]. The World Health Organization currently lists mental health as one of the main health issues for adolescents globally [3]. Consequently, mental well-being is defined by more than the mere absence of mental illness. Historically, the fields of epidemiology, mental health research, and practice, have been predominantly focused on pathological aspects of mental health. There is an imperative for a paradigm shift that emphasizes the positive dimensions of health [4,5].

### Measuring mental well-being

Mental health encompasses both positive and negative dimensions [6–8] and it is important to note that mental well-being can coexist with mental illness [9]. Therefore, mental well-being should be studied in its own right, not just as absence of mental illness [10].

There are different ways to define and operationalise mental well-being [11,12], but it is often defined as a multidimensional concept that includes emotional well-being, or hedonism, and functional well-being, or eudaimonism [13]. One widespread operationalisation based on this definition is the Mental Health Continuum [14]. Here, mental well-being is regarded as a set of symptoms that describes a person's mental well-being within three dimensions: emotional (corresponding to hedonism), psychological and social (together corresponding to eudaimonism) [14].

High levels of mental well-being are associated with cognitive, physical, and social health [10,15–17]. Further, longitudinal studies show that individuals with high mental well-being have a lower risk of the most common mental disorders in a long-term perspective [14,18,19]. These disorders commonly emerge in adolescence [20]. The levels of mental well-being in adolescence are therefore of great importance.

### Distribution of mental well-being in the population

From a public health perspective, not only the overall levels of health and well-being are of importance, their distribution matters as well. Social determinants, such as the conditions in which people live and the health systems they can access, have considerable effects on health outcomes [21], including mental health.

Little is known about the distribution of mental well-being among adolescents. Most studies have either examined the distribution of mental illness or addressed well-being in ways that did not include both hedonic and eudaimonic well-being. Nevertheless, existing studies show that socioeconomically disadvantaged children and adolescents are two to three times more likely to develop mental health problems [22]. Further, there are significant sex differences in mental well-being and mental health problems among children and adolescents, with girls reporting lower levels of

well-being and higher levels of mental health problems than boys [23,24]. Adolescents who identify as non-heterosexual experience lower levels of mental health than their heterosexual peers [25,26]. There is also an association between disability and mental health, where adolescents with sensory (visual and hearing) impairments report more mental health problems than their peers without impairments [27–29]. Likewise, neurodevelopmental disorder (NDD) seems to be connected to mental health, as adolescents with NDD experience more mental health problems than adolescents without NDD, an association especially strong for girls [28,30].

The importance of gaining knowledge of the health distribution in the population was pointed out by the Commission on Social Determinants [31]. Due to the sparseness of research, there is a need to examine associations between social determinants and mental well-being among adolescents.

### Temporal changes in mental well-being

Not only is there a lack of research on social determinants and mental well-being, but research regarding changes over time in adolescents' mental well-being is also scarce. However, studies show an overall decrease in mental well-being in most of the countries investigated. A study of positive mental health (operationalised as self-rated health, life satisfaction and multiple health complaints) among adolescents in the Nordic countries 2002–2014 showed negative development in Denmark, Iceland, Finland and Sweden, while Norway showed a positive development [32]. Another study showed a decrease in mental well-being (operationalised as emotional and functional well-being) for Scottish girls 2003–2014 [33]. From an epidemiological perspective, it is important to monitor the development over time, as changes in mental health are important for public health.

### The current study

There is a lack of high-quality population-based data on adolescent mental well-being, for different groups and over time. This kind of data is important for public health policy and promotion. The aim of this study was to examine the level and distribution of mental well-being as well as changes over time in a Swedish adolescent population, using a validated measure that includes both hedonic and eudemonic well-being. To fulfil this aim, the following research questions were addressed: 1) What is the level of mental well-being in the population? 2) Are there differences in levels of mental well-being in different social groups, and if so, how large are they? 3) Are there differences in levels of mental well-being in different social groups over time? 4) Can variations in levels of mental well-being be explained by different group affiliations, and if so, by how much?

## Materials and methods

### Study design, population and data collection

To fulfil the overall aim and answer the research questions we used a quantitative approach based on four waves (2014, 2017, 2020 and 2023) of the recurrent cross-sectional whole population survey Adolescent Life and Health. This is a comprehensive health survey about living conditions, life habits and various aspects of health including scales of mental well-being, conducted in Västmanland county in Sweden. Västmanland County is one of Sweden's 21 counties. Compared to the Swedish average, Västmanland exhibits a somewhat lower proportion of post-secondary education (Västmanland = 40.8%, Sweden = 46.9% (2023)), a somewhat higher unemployment rate (Västmanland = 4.2%, Sweden = 3.4% (2023)), and a higher proportion of young inhabitants living with a low economic standard (Västmanland = 21%, Sweden = 17% (2022)) [34]. In terms of representativeness, Västmanland can therefore be considered slightly more disadvantaged than Sweden as a whole.

The population in this study consisted of students in ninth grade of elementary school (~15 years) and second grade of upper secondary school (~17 years) in all schools in Västmanland county. Students within special needs education and

immigrant students in language introductory classes were excluded from participation. The survey was conducted during school hours under exam-like conditions in a classroom context, in 2014 as a paper survey and the following years as a digital survey. Students received oral and written information about the purpose and content of the survey, that participation was voluntary and would be anonymous, i.e., no names or personal identification number would be collected, that questions could be skipped, and that participation could be cancelled at any time for any reason. The students were also informed that consent to participate was given by completing the questionnaire. This so-called implied informed consent was deemed sufficient since the survey was completely anonymous. Since the survey was anonymous, no requirement for ethical approval under Swedish law (Ethical Review Act 2003:460) is required. Nevertheless, to have the latter confirmed, an ethical vetting was carried out for each survey wave. The Swedish Ethical Review Authority responded that the studies did not need formal approval but gave – when relevant – advisory opinions to their decisions (Dnr 2013/464, Dnr 2016/480, Dnr 2019–05620, Dnr 2022-05672-01). With respected to participants who were minors, their custodians were provided with prior information regarding the study. Custodians with a child younger than 15 years were given the option to inform the school if they did not wish for their child to participate.

### Outcome variable: Level of mental well-being

Mental well-being was measured with the Mental Health Continuum Short Form (MHC-SF) [35], covering both hedonic and eudaimonic aspects of well-being. In MHC-SF, hedonic well-being corresponds to Emotional well-being, which consists of three indicators: perceived life satisfaction, happiness, and interest in life. Eudaimonic – or functional – well-being corresponds to Social and Psychological well-being. Social well-being consists of five dimensions: social acceptance, social actualisation, social coherence, social contribution, and social integration. Psychological well-being consists of six dimensions: autonomy, environmental mastery, self-acceptance, personal growth, and purpose in life. In total, MHC-SF consists of 14 items. The respondents report how often they experience each of the 14 items on a scale from 0 (Never) to 5 (Every day). By totalling the score of all items, a continuous outcome variable can be generated ranging 0–70, a higher score reflects higher mental well-being.

The MHC-SF continuous scale has undergone psychometric evaluation in a number of international studies on adolescents [36–39], as well as in Sweden [40]. The results of the Swedish study indicated that the MHC-SF, when administered to Swedish adolescents, manifests as essentially unidimensional, with a predominant general well-being factor and high internal reliability (coefficient omega = 0.88).

### Explanatory factors

Ten explanatory factors were chosen to examine differences in mental well-being in the study population. All except socioeconomic status are associated with the discriminatory factors in the Swedish Discrimination Act (Law 2008:567). Additionally, though socioeconomic status is not considered a discriminatory factor, it is important when it comes to health equality [41] and thus included in the study. Each of these factors and how they were measured are described below.

**Grade.** Each survey form was pre-coded with school and grade. This is the only explanatory factor variable that is not based on the student's response.

**Sex.** The Sex-variable was self-reported by the students in the survey, answering the question "Are you…?" with the alternatives "Boy" or "Girl". 2017–2023 there was a third alternative; "Other". Since this category was missing in 2014 and because the low number of respondents in this category, the category was not reported in the results of this study. However, the individuals are included in the total of the study.

**Sexual orientation.** The students in the survey were asked to indicate their sexual orientation by answering the questions "Which of the following concepts do you think describes you?" (2014) and "With which sexual orientation do you identify?" (2017–2023) with the alternatives "Heterosexual", "Bisexual", "Homosexual", "Other", and "Insecure". In the analysis, the answers were dichotomised into "Heterosexual" and "Non-heterosexual", due to small number in the sub-categories in the latter.

**Socioeconomic status.** Socioeconomic status was represented by one self-reported item adapted from the MacArthur Scale of Subjective Social Status (SSES) – Youth Version [42]. Respondents were asked to answer the following question on a scale of 1 (lowest = lowest status) to 7 (highest = highest status): "Imagine society as a ladder. If you think about your family´s finances in comparison with society at large, where would you put your family on the ladder?" Due to small numbers in the two lowest categories, they were merged in the analysis. Research has demonstrated a robust correlation between adolescents´ subjective perceptions of socioeconomic status, as measured by the society ladder, and their mental health [43].

**Country of birth.** To determine country of birth, the students were asked to answer the question "Where were you born" with the response alternatives "Sweden", "Rest of Europe" or "Rest of the world".

### Visual, hearing or mobility impairment, specific learning disorder and neurodevelopmental disorder

The students were asked to answer a multiple-choice question with Yes or No if they had any of the following disabilities:

- Hearing impairment

- Visual impairment where eyeglasses or contact lenses are insufficient

- Mobility impairment

- Reading/writing difficulties, dyslexia, or dyscalculia (hereafter referred to as Specific learning disorder (SLD))

- Autism Spectrum Disorder, Attention-Deficit/Hyperactivity Disorder and Tic Disorders (hereafter referred to as Neurode-velopmental disorder (NDD)).

### Statistical analysis

The present study only contains survey questions available for all four waves. To reduce the risk of unreliable responses, spam filters were applied in all four waves to rule out, e.g., extreme combinations or response patterns; in 2014 they took the form of external reviewers, and in 2017–2023 computerised filters were used.

To answer the first research question, descriptive population parameters (unadjusted means, standard deviations, and proportions) were produced. Crude mean values on the continuous MHC-SF-scale were calculated for the overall population and for different sub-groups in the study population.

To answer the second research question, effect sizes per survey year and totals are presented for each explanatory factor. Independent-samples two-tailed t-tests were conducted to examine cross-sectional differences in level of mental well-being for factors with two categories, with Cohen's $d$ to assess effect sizes, presented as point estimates. One-way ANOVA was performed to compute effect sizes for factors with more than two categories. In social science data, the recommended minimum level for effect size estimates to assess practically significant effect is 0.41 for Cohen's $d$ and 0.04 for Eta squared ($\eta^2$) [44]. P-values are presented for each factor and survey year as well as for the total.

To test changes over time, the third research question was answered by a trend analysis through Reverse Helmert's Contrast [45]. An advantage of this method is that it contrasts the mean score for one time point with the average mean score for the previous time points, which gives a robust estimation. In the current study the 2017 mean score was contrasted with the 2014 mean score; the 2020 mean score with the average mean scores for 2014 and 2017; and the 2023 mean score with the average mean scores for 2014–2020. Contrasts with p-values < 0,05 were considered statistically significant. In addition, error bars were produced for the continuous mental well-being variable, with two purposes: first, to visualise the mean level of mental well-being with confidence intervals for each explanatory factor and category; second, to assess whether the between-group difference motivated the use of interaction terms in the regression analysis.

The fourth research question was answered through a hierarchical multiple linear regression analysis to further assess the explanatory factors' associations with the dependent variable, the level of mental well-being (MHC-SF continuous scale). The univariate analysis was used to determine the explanatory factors to be included in the analysis. Based on Ferguson´s [44] recommended minimum effect size for social science data, factors that showed no or very small significant effect ($d < 0.41$, $\eta^2 < 0.04$) on mental well-being (grade, country of birth, visual impairment, and SLD) were excluded. Hierarchical regression was performed in blocks to elucidate the contributions of each block and category of explanatory factors contributed to explaining the variation in mental well-being. Assumptions underlying the robustness of the analysis (e.g., normality of residuals and no perfect multicollinearity) were confirmed, resulting in a final regression consisting of seven blocks, the first six of which included main effects and the seventh addressed interaction effects. The independent variables were entered in the following order: 1) Sex; 2) Sexual orientation; 3) NDD; 4) Hearing impairment and Mobility impairment; 5) SSES; 6) Survey year; and 7) Interaction terms for each dichotomous variables×survey year and SSES×survey year. Multicollinearity was controlled for through the examination of the Variance Inflation Factor (VIF) values. The VIF-values for the main effects were <10 (range 1.00–1.03), indicating no multicollinearity in the data. The VIF-values were higher for some variables in the seventh block, as can be expected when interaction terms are introduced.

All analyses were conducted using statistical software SPSS v.29.

## Results

### Description of sample

In total, 16 288 adolescents participated corresponding to a total response rate of 75%, ranging from 72% in 2020 to 78% in 2014. Missing data for the outcome was 7.9%. Missing data per item in MHC-SF varied from 3.1–4.9%. If data were missing for the outcome, subjects were removed from the study. For the explanatory variables, missing data ranged from 2.9–7%, except for Sexual orientation (10.7%). If data were missing for one or more explanatory factors but were available for the outcome, individuals were included in the total of the descriptive analysis of mental well-being. In the multivariate analysis, subjects were excluded listwise from analysis if data were missing for one or more explanatory factors.

Table 1 displays sample characteristics over the study period. In total, 50% of the population was in 9th grade of elementary school and 50% in 2nd grade of upper secondary school (mean age: 2014 = 16.20; 2017 = 16.26; 2020 = 16.30; 2023 = 16.15). The sex distribution was 50/50. The majority was born in Sweden and a small but increasing proportion was born outside of Europe. The vast majority, 88%, identified themselves as heterosexual, but the four survey waves display a growing number of non-heterosexual adolescents. The proportion that assessed their SSES as 1, 2, 3, or 7 was stable over time, whilst the proportion that assessed their SSES as 4 was increasing and 5 and 6 was decreasing. Hearing impairment was decreasing, visual and mobility impairment were stable over time and NDD and SLD were increasing.

### Level of mental well-being, differences between social groups and changes over time

Table 2 describes the mean values for MHC-SF continuous scale and changes over the study period. For the overall study population, the mean level of mental well-being was 42.92, with a range of 42.27–43.98 over the four survey waves. Mental well-being decreased significantly during the study period, from 43.98 in 2014 to 42.27 in 2023. The reverse Helmert's contrasts yield significant decreasing mean levels between all study waves.

For 9th grade, there is a significant decrease in 2017 and in 2023 compared to the mean level of mental well-being for the previous study periods. For 2nd grade the mean levels for 2020 and 2023 show a significant decrease compared to the previous study periods.

The mean mental well-being score for girls is 40.73 (range 39.35–42.36) and for boys 45.41 (range 44.93–45.79). A statistically significant decrease in level of mental well-being is seen for girls during the study period, from 42.36 in 2014

**Table 1. Sample descriptives per group and year.**

| | | 2014 | | 2017 | | 2020 | | 2023 | | Total | |
|---|---|---|---|---|---|---|---|---|---|---|---|
| | | N | % | N | % | N | % | N | % | N | % |
| **Grade** | 9th grade | 2087 | 52% | 1984 | 50% | 2001 | 51% | 2317 | 53% | 8389 | 52% |
| | 2nd grade | 1958 | 48% | 1966 | 50% | 1953 | 49% | 2022 | 47% | 7899 | 49% |
| **Sex** | Girl | 2066 | 52% | 1887 | 50% | 1950 | 50% | 2098 | 50% | 8001 | 50% |
| | Boy | 1943 | 49% | 1877 | 50% | 1930 | 50% | 2118 | 50% | 7868 | 50% |
| **Sexual orientation** | Non-heterosexual | 351 | 10% | 391 | 12% | 488 | 15% | 594 | 14% | 1824 | 13% |
| | Heterosexual | 3284 | 90% | 2944 | 88% | 2813 | 85% | 3687 | 86% | 12728 | 88% |
| **Subjective socioeconomic status (SSES)\*** | 1 (Lowest/poorest) | 41 | 1% | 32 | 1% | 52 | 1% | 35 | 1% | 160 | 1% |
| | 2 | 121 | 3% | 76 | 2% | 89 | 2% | 79 | 2% | 365 | 2% |
| | 3 | 357 | 9% | 280 | 9% | 335 | 9% | 417 | 10% | 1389 | 9% |
| | 4 | 1080 | 27% | 928 | 28% | 1163 | 31% | 1399 | 35% | 4570 | 30% |
| | 5 | 1652 | 42% | 1357 | 41% | 1613 | 42% | 1600 | 39% | 6222 | 41% |
| | 6 | 627 | 16% | 458 | 14% | 410 | 11% | 392 | 10% | 1887 | 13% |
| | 7 (Highest/richest) | 107 | 3% | 164 | 5% | 154 | 4% | 134 | 3% | 559 | 4% |
| **Country of birth** | Sweden | 3536 | 88% | 3251 | 86% | 3175 | 82% | 3523 | 82% | 13485 | 84% |
| | Rest of Europe | 121 | 3% | 107 | 3% | 128 | 3% | 140 | 3% | 496 | 3% |
| | Rest of the world | 349 | 9% | 444 | 12% | 588 | 15% | 619 | 15% | 2000 | 13% |
| **Hearing impairment** | No | 3758 | 93% | 3262 | 93% | 3600 | 95% | 4126 | 97% | 14746 | 95% |
| | Yes | 287 | 7% | 234 | 7% | 186 | 5% | 126 | 3% | 833 | 5% |
| **Visual impairment** | No | 3811 | 94% | 3262 | 94% | 3544 | 94% | 4007 | 94% | 14624 | 94% |
| | Yes | 234 | 6% | 222 | 6% | 245 | 7% | 252 | 6% | 953 | 6% |
| **Mobility impairment** | No | 3929 | 97% | 3353 | 97% | 3684 | 98% | 4178 | 98% | 15144 | 98% |
| | Yes | 116 | 3% | 99 | 3% | 92 | 2% | 73 | 2% | 380 | 2% |
| **Neurodevelopmental disorder (NDD)** | No | 3716 | 92% | 3168 | 89% | 3388 | 89% | 3685 | 87% | 13957 | 89% |
| | Yes | 329 | 8% | 395 | 11% | 425 | 11% | 575 | 14% | 1724 | 11% |
| **Specific Learning Disorders (SLD)** | No | 3676 | 91% | 3004 | 85% | 3275 | 86% | 3700 | 87% | 13655 | 87% |
| | Yes | 369 | 9% | 528 | 15% | 526 | 14% | 562 | 13% | 1985 | 13% |

\*Due to small numbers, the first two categories are merged in the analysis.

to 39.35 in 2023, with significant decline in 2017 and 2023 compared to the means of the previous periods. No significant changes are seen for boys during the study period. The effect size of the difference in level of mental well-being between girls and boys is $d = -0.328$ (range $d = -0.250 - -0.439$ for each study wave).

For non-heterosexual adolescents, the level of mental well-being was 36.03 (range 34.52–37.38). There was a significant decrease in level of mental well-being in 2017 (34.52) compared to 2014 (37.38). The effect size of the overall difference between non-heterosexual and heterosexual adolescents was a standardised mean difference of $d = -0.548$ (range $d = -0.484 - -0.642$).

Fig 1 displays the distribution of levels of mental well-being in different SSES-groups by survey year. This shows an almost linear association between SSES and level of mental well-being: the higher the SSES, the higher the mental well-being. The lowest SSES group had a mean mental well-being score of 36.72 (range 34.41–37.98), compared to the highest SSES group that had a mean score of 48.18 (range 46.34–51.83). The effect size of the overall difference in level of mental well-being between SSES-groups was $\eta^2 = 0.051$ (range 0.048–0.058). There were few significant changes over time for the different SSES-groups.

**Table 2. Level of mental well-being (mean and standard deviation) with effect size and trend analysis by different groups.**

| | | 2014 | 2017 | 2020 | 2023 | Total |
|---|---|---|---|---|---|---|
| **Overall study population** | | 43.98 (SD 13.90) | 42.74 (SD 14.86) ˅ | 42.69 (SD 14.78) ˅ | 42.27 (SD 14.64) ˅ | 42.92 (SD 14.56) |
| **Grade** | 9th grade | 44.15 (SD 14.13) | 42.52 (SD 15.41) ˅ | 42.84 (SD 15.44) | 42.34 (SD 14.99) ˅ | 42.96 (SD 15.00) |
| | 2nd grade | 43.81 (SD 13.65) | 42.94 (SD 14.32) | 42.53 (SD 14.12) ˅ | 42.20 (SD 14.24) ˅ | 42.87 (SD 14.09) |
| | Effect size (Cohen's d) | 0.024 (CI -0.039– 0.088, p 0.457) | -0.028 (CI -0.095– 0.038, p 0.403) | 0.021 (CI -0.044– 0.085, p 0.531) | 0.009 (CI -0.053– 0.071, p 0.766) | 0.007 (CI -0.025– 0.039, p 0.684) |
| **Sex** | Girl | 42.36 (SD 13.36) | 40.50 (SD 14.25) ˅ | 40.66 (SD 14.65) | 39.35 (SD 13.88) ˅ | 40.73 (SD 14.07) |
| | Boy | 45.79 (SD 14.16) | 45.31 (SD 14.69) | 44.93 (SD 14.51) | 45.57 (SD 14.39) | 45.41 (SD 14.43) |
| | Effect size (Cohen's d) | -0.250 (CI -0.314 – -0.185, p<0.001) | -0.332 (CI -0.400 – -0.264, p<0.001) | -0.293 (CI -0.359 – -0.227, p<0.001) | **-0.439** (CI -0.503 – -0.376, p<0.001) | -0.328 (CI -0.361 – -0.296, p<0.001) |
| **Sexual orientation** | Non-heterosexual | 37.38 (SD 15.39) | 34.52 (SD 15.50) ˅ | 36.67 (SD 15.77) | 35.67 (SD 15.50) | 36.03 (SD 15.57) |
| | Heterosexual | 44.62 (SD 13.58) | 43.77 (SD 14.26) ˅ | 43.64 (SD 14.16) | 43.32 (SD 14.20) ˅ | 43.83 (SD 14.05) |
| | Effect size (Cohen's d) | **-0.526** (CI -0.642– -0.411, p<0.001) | **-0.642** (CI -0.752– -0.532, p<0.001) | **-0.484** (CI -0.582– -0.386, p<0.001) | **-0.532** (CI -0.625– -0.440, p<0.001) | **-0.548** (CI -0.599– -0.496, p<0.001) |
| **Subjective socioeconomic status (SSES)*** | 1+2 | 37.02 (SD 15.18) | 34.41 (SD 15.58) | 37.12 (SD 16.19) | 37.98 (SD 18.02) | 36.72 (SD 16.18) |
| | 3 | 38.64 (SD 13.67) | 35.91 (SD 14.94) ˅ | 36.04 (SD 15.10) | 36.40 (SD 14.29) | 36.79 (SD 14.49) |
| | 4 | 41.90 (SD 13.57) | 41.88 (SD 14.23) | 40.47 (SD 14.48) ˅ | 39.81 (SD 14.06) ˅ | 40.90 (SD 14.12) |
| | 5 | 45.03 (SD 13.60) | 44.29 (SD 13.70) | 44.69 (SD 13.79) | 44.28 (SD 13.12) | 44.59 (SD 13.55) |
| | 6 | 48.25 (SD 12.26) | 46.91 (SD 13.58) | 46.91 (SD 13.44) | 47.56 (SD 14.18) | 47.49 (SD 13.25) |
| | 7 | 51.83 (SD 14.73) | 46.65 (SD 19.16) ˅ | 46.34 (SD 19.56) | 49.11 (SD 17.24) | 48.18 (SD 18.10) |
| | Effect size (Eta-squared) | **0.055** (CI 0.041– 0.069, p<0.001) | **0.048** (CI 0.033– 0.062, p<0.001) | **0.049** (CI 0.035– 0.062, p<0.001) | **0.058** (CI 0.043– 0.072, p<0.001) | **0.051** (CI 0.044– 0.058, p<0.001) |
| **Country of birth** | Sweden | 43.91 (SD 13.91) | 42.90 (SD 14.64) ˅ | 42.64 (SD 14.57) ˅ | 42.08 (SD 14.15) ˅ | 42.89 (SD 14.32) |
| | Rest of Europe | 42.24 (SD 15.13) | 42.05 (SD 16.06) | 43.71 (SD 14.37) | 41.48 (SD 16.44) | 42.37 (SD 15.49) |
| | Rest of the world | 45.80 (SD 12.73) | 42.52 (SD 15.35) ˅ | 43.15 (SD 15.81) | 43.89 (SD 16.18) | 43.73 (SD 15.34) |
| | Effect size (Eta-squared) | 0.002 (CI 0.000– 0.005, p 0.026) | 0.000 (CI 0.000– 0.001, p 0.782) | 0.000 (CI 0.000– 0.002, p 0.595) | 0.002 (CI 0.000– 0.005, p 0.021) | 0.000 (CI 0.000– 0.001, p 0.052) |
| **Hearing impairment** | No | 44.08 (SD 13.91) | 43.41 (SD 14.69) | 43.14 (SD 14.61) ˅ | 42.50 (SD 14.55) ˅ | 43.26 (SD 14.44) |
| | Yes | 42.73 (SD 13.65) | 39.52 (SD 14.51) ˅ | 35.53 (SD 15.75) ˅ | 36.61 (SD 16.73) | 39.28 (SD 15.13) |
| | Effect size (Cohen's d) | 0.097 (CI -0.029– 0.222, p 0.131) | 0.265 (CI 0.121– 0.409, p<0.001) | **0.519** (CI 0364. – 0.673, p<0.001) | 0.403 (CI 0.220– 0.586, p<0.001) | 0.275 (CI 0.201– 0.348, p<0.001) |
| **Visual impairment** | No | 44.00 (SD 13.95) | 43.47 (SD 14.54) | 42.98 (SD 14.62) ˅ | 42.37 (SD 14.53) ˅ | 43.19 (SD 14.41) |
| | Yes | 43.76 (SD 12.92) | 38.17 (SD 16.70) ˅ | 39.45 (SD 16.27) | 41.42 (SD 16.42) | 40.75 (SD 15.77) |
| | Effect size (Cohen's d) | 0.017 (CI -0.122– 0.157, p 0.808) | 0.361 (CI 0. – 0., p<0.001) | 0.240 (CI 0. – 0., p 0.002) | 0.065 (CI 0. – 0., p 0.339) | 0.168 (CI 0.099– 0.237, p<0.001) |
| **Mobility impairment** | No | 43.98 (SD 13.93) | 43.38 (SD 14.67) | 42.99 (SD 14.64) ˅ | 42.44 (SD 14.53) ˅ | 43.18 (SD 14.44) |
| | Yes | 44.22 (SD 12.68) | 35.01 (SD 14.07) ˅ | 32.66 (SD 15.92) ˅ | 35.12 (SD 19.25) | 37.26 (SD 15.94) |
| | Effect size (Cohen's d) | -0.017 (CI -0.214– 0.180, p 0.864) | **0.571** (CI 0.354– 0.788, p<0.001) | **0.704** (CI 0.486– 0.922, p<0.001) | **0.500** (CI 0.255– 0.746, p 0.003) | 0.409 (CI 0.300– 0.518, p<0.001) |
| **Neuro-developmental disorder (NDD)** | No | 44.48 (SD 13.72) | 43.73 (SD 14.50) ˅ | 43.53 (SD 14.39) | 42.87 (SD 14.50) ˅ | 43.66 (SD 14.28) |
| | Yes | 38.05 (SD 14.65) | 37.04 (SD 15.27) | 35.87 (SD 15.79) | 38.64 (SD 14.95) ^ | 37.48 (SD 15.21) |
| | Effect size (Cohen's d) | **0.466** (CI 0.347– 0.586, p<0.001) | **0.458** (CI 0.346– 0.571, p<0.001) | **0.526** (CI 0.421– 0.632, p<0.001) | 0.291 (CI 0.199– 0.382, p<0.001) | **0.430** (CI 0.377– 0.482, p<0.001) |
| **Specific Learning Disorders (SLD)** | No | 44.08 (SD 13.91) | 43.74 (SD 14.47) | 43.13(SD 14.64) ˅ | 42.58 (SD 14.48) ˅ | 43.37 (SD 14.37) |
| | Yes | 42.95 (SD 13.80) | 39.43 (SD 15.64) ˅ | 40.25 (SD 15.23) | 40.46 (SD 15.69) | 40.61 (SD 15.24) |
| | Effect size (Cohen's d) | 0.082 (CI -0.030– 0.194, p 0.152) | 0.295 (CI 0.195– 0.394, p<0.001) | 0.195 (CI 0.099– 0.291, p<0.001) | 0.145 (CI 0.051– 0.239, p 0.005) | 0.191 (CI 0.141– 0.240, p<0.001) |

Effect sizes above recommended minimum level for practically significant effect (d>0.41; η²>0.04) marked in **bold**.

^ = Significant (p<0,05) incline of the mean compared to the mean from previous survey waves.

˅ = Significant (p<0,05) decline of the mean compared to the mean from previous survey waves.

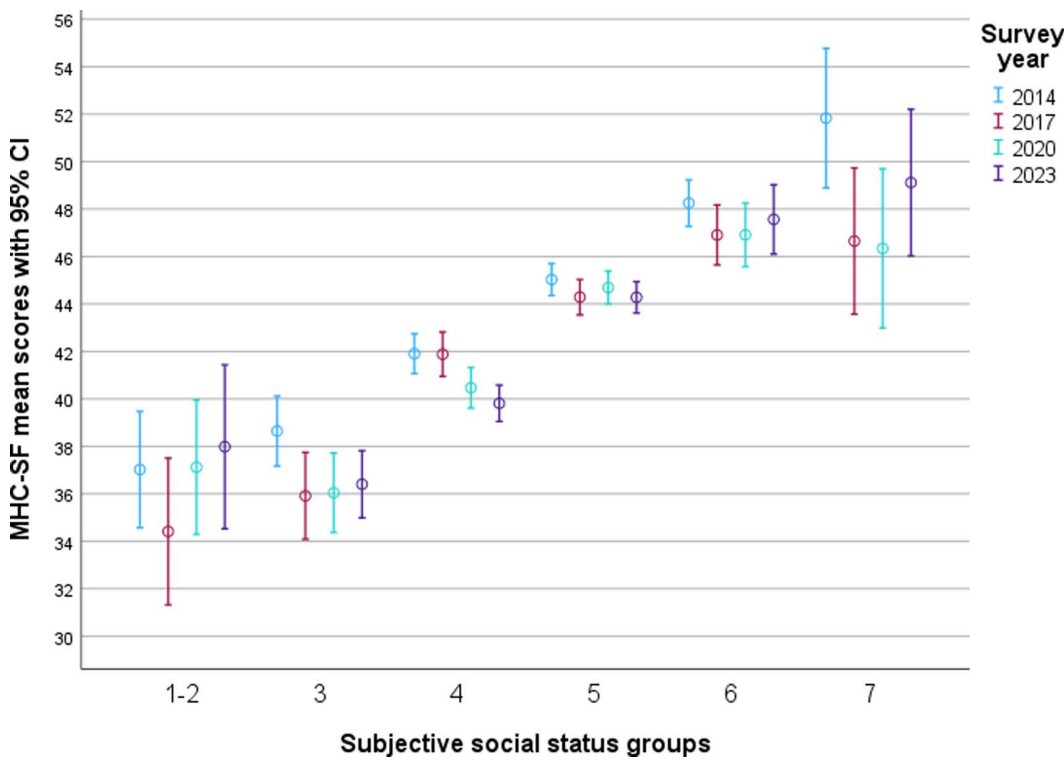

**Fig 1. Distribution of mental well-being by SSES-groups and year.** Note the broken scale.

For adolescents born outside of Europe there was a significant decline in level of mental well-being 2014–2017. Adolescents born outside of Europe had significantly higher mean values of mental well-being in 2014 and in 2023 compared to adolescents born in Sweden. No other significant differences were seen between countries of birth.

The mean mental well-being score for adolescents with hearing impairment was 39.28 (range 35.53–42.73), for visual impairment 40.75 (range 38.17–43.76) and for mobility impairment 37.26 (range 32.66–44.22). For adolescents with hearing impairment or mobility impairment, there was a significant decrease in level of mental well-being in 2017 and 2020 compared to the previous periods. For visual impairment, a significant decrease was seen only 2017. The difference in level of mental well-being between adolescents with and without impairment in terms of effect size was $d=0.275$ (range 0.097–0.519) for hearing impairment, $d=0.168$ (range 0.017–0.361) for visual impairment, and $d=0.409$ (range -0.017–0.704) for mobility impairment.

For adolescents with NDD, the mean mental well-being score was 37.48 (range 35.87–38.64). An increase in level of mental well-being was seen in 2023 compared to the previous periods. The effect size compared to adolescents without NDD was $d=0.430$ (range $d=0.291$–0.526). For adolescents with SLD the mean score was 40.61 (range 39.43–42.95). A significant decrease was seen in level of mental well-being 2017 (39.43) compared to 2014 (42.95). Effect sizes for adolescents with SLD were $d=0.191$ (range 0.082–0.295).

### The role of the explanatory factors

In the regression analysis, mobility impairment showed no significance and was thus excluded in the final model. All interaction terms except *sex×survey year* and *hearing impairment×survey year* yielded non-significant associations, therefore only these two interaction terms were included in the final model. Consequently, the final model consisted of six blocks of main effects and a seventh block with interaction terms.

**Table 3. Model statistics and standardized coefficients of main effects and interaction effects.**

| | Block 1 | Block 2 | Block 3 | Block 4 | Block 5 | Block 6 | Block 7 |
|---|---|---|---|---|---|---|---|
| **Model statistics (N = 12524)** | | | | | | | |
| Model p-value | <0.001 | <0.001 | <0.001 | <0.001 | <0.001 | <0.001 | <0.001 |
| R | 0.173 | 0.226 | 0.255 | 0.258 | 0.329 | 0.330 | 0.332 |
| $R^2$ | 0.030 | 0.051 | 0.065 | 0.067 | 0.108 | 0.109 | 0.110 |
| $\Delta R^2$ | 0.030 | 0.021 | 0.014 | 0.001 | 0.041 | 0.001 | 0.002 |
| P-value F-change | <0.001 | <0.001 | <0.001 | <0.001 | <0.001 | 0.002 | <0.001 |
| **Main effects (standardized coefficients, β)** | | | | | | | |
| Sex[a] | 0.173 *** | 0.157 *** | 0.163 *** | 0.163 *** | 0.149 *** | 0.150 *** | 0.077 *** |
| Sexual orientation[b] | | 0.147 *** | 0.136 *** | 0.135 *** | 0.124 *** | 0.123 *** | 0.121 *** |
| NDD[c] | | | -0.119 *** | -0.114 *** | -0.100 *** | -0.098 *** | -0.098 *** |
| Hearing impairment[d] | | | | -0.037 *** | -0.033 *** | -0.035 *** | n.s. |
| SSES | | | | | 0.205 *** | 0.204 *** | 0.204 *** |
| Survey year | | | | | | -0.026 ** | -0.122 *** |
| **Interaction effects (standardized coefficients, β)** | | | | | | | |
| Sex × survey year | | | | | | | 0.130 *** |
| Hearing impairment × survey year | | | | | | | -0.048 ** |

Reference category: a=Girl; b=non-heterosexual; c=no NDD; d=no hearing impairment

***=p<0.001, **=p<0.05, n.s=non-significant

Overall, all the blocks significantly improved the model (Table 3) (for more detailed coefficients, see S1 Table). The blocks that added most to the explained variance were the first block that included sex (ΔR2 = 0.03), the second block that included sexual orientation (ΔR2 = 0.021), the third block that included NDD and SLD (ΔR2 = 0.014) and the fifth block that included SSES (ΔR2 = 0.041). The full model with all main effects (block 6) explained 10.9% of the variation in mental well-being among adolescents in the study population. The standardised beta coefficients in block six showed that the factor that explained most variation in mental well-being was SSES (β = 0.204), followed by sex (β = 0.150) and sexual orientation (β = 0.123).

The final model (block 7) including all explanatory factors as well as interaction terms explained 11% of the variation in the outcome. Fig 2 shows the mean level of mental well-being and temporal trends for girls and boys respectively, with increased sex differences over time. The standardised beta coefficient for the interaction of sex and survey year (β = 0.130) indicated that the increased sex difference in mental well-being over time partly explains the variation, and that the combined effect of sex and survey year explains more variation than the two explanatory factors do separately.

## Discussion

The current study examines the level, distribution, and temporal changes of mental well-being in a Swedish adolescent population. Previous research on the epidemiology of adolescent mental health, with a focus on temporal variations and differences between social groups, has predominantly focused mainly on mental health problems rather than well-being. However, the concept of well-being possesses its own inherent value, and it is therefore important to explore and monitor mental well-being as a distinct entity. This study is, to the best of our knowledge, the first to do so from an equality perspective, based on whole-population data, with a validated instrument that measures both hedonic and eudaimonic well-being.

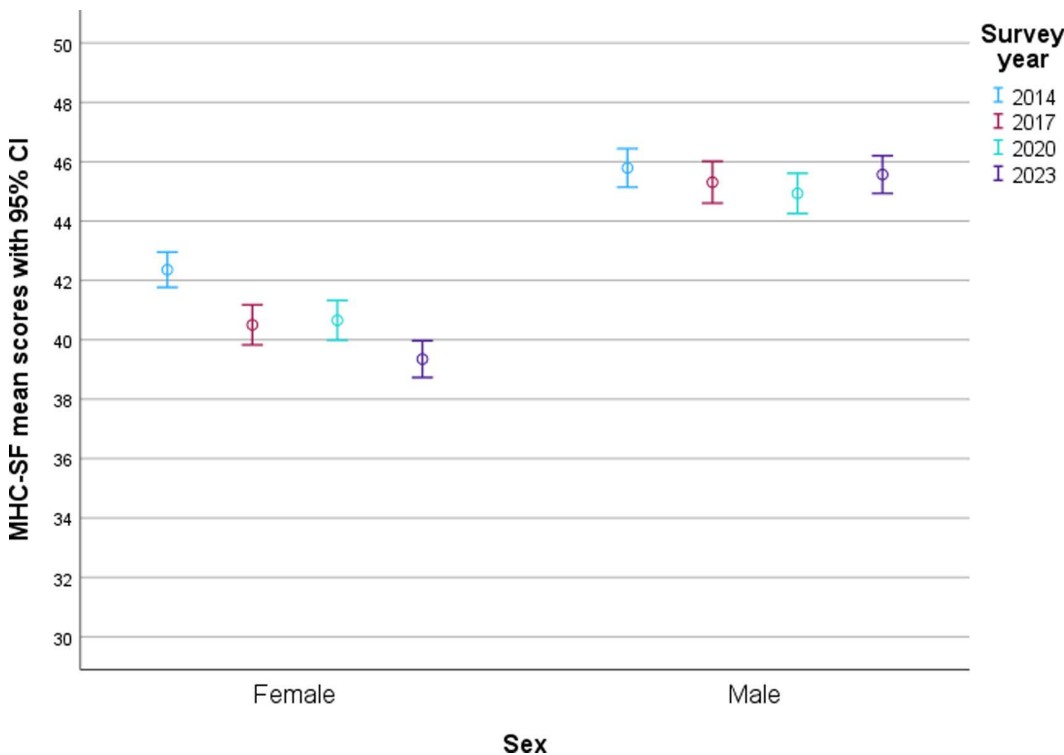

**Fig 2. Error bar of MHC-SF mean score and 95% confidence interval by sex and survey year.** Note the broken scale.

The results of reverse Helmert's contrast indicate that the overall level of mental well-being has decreased significantly over the 9-year study period. However, further analysis employing hierarchical multiple regression, including interaction terms, shows that this deterioration is predominantly attributed to the decrease observed among girls, and to some extent among adolescents with hearing impairment.

Significant cross-sectional differences in the level of mental well-being based on sex was observed, with boys reporting higher levels of mental well-being. This pattern appears to be reinforced over time. In addition, non-normative groups report lower mental well-being compared to normative groups; non-heterosexual adolescents report lower levels compared to heterosexual adolescents, as do adolescents with impairments compared to adolescents without impairments. Furthermore, adolescents with low subjective socioeconomic status report lower mental well-being compared to their high-status peers. SSES emerges as the predominant factor explaining inter-group disparities in mental well-being, followed by factors such as sex and sexual orientation.

The mean level of mental well-being in the study population was 42.92 (range of 42.27–43.98) on the 0–70 scale over the study period. With respect to generalisability, it is conceivable that the level of mental well-being in the current study could be a slight underestimate compared to Sweden as a whole, as Västmanland as a region can be regarded as having a marginally lower socioeconomic level than Sweden as a whole. However, this is not expected to affect the relative differences (i.e., disparities) between different groups in the results. Although the MHC-SF is a widely used and well-validated instrument [36–40], there are few studies to compare this result with. One study that made a cross-cultural comparison with slightly older youths (age 16–25) in Denmark, Canada and the Netherlands found a mean MHC-SF score ranging from 44.2 (the Netherlands) to 55.7 (Canada) [36]. Compared to these results, the current study highlights a promotive potential, as the results in this study are lower. However, since the former data are from 2013–2016,

the temporal factor needs to be taken into consideration as the results in these countries could have changed since the studies were conducted.

The explanatory factors with largest effect sizes between groups are sexual orientation and SSES, that both meet Ferguson's [44] recommended minimum effect size for practical significance for all study waves. Effect sizes for Mobility impairment and NDD are also above the threshold for three of the measuring points. However, it is important to note that none of the effect sizes can be considered as having moderate or strong effect. In the context of sex, the effect size only exceeds the threshold for practical significance for 2023, aligning with the regression analysis which demonstrates an increasing disparity between the sexes. While the effect size is small, it is relevant to note that this disparity pertains to half of the population, emphasising the potential for significant impact on public health.

The overall effect size of the final regression model ($R = 0.332$) exceeds the threshold suggested by Ferguson [44], yet it is not considered to be moderate or strong. The final regression model accounts for a mere 11% of the observed variation in mental well-being, suggesting that there are other factors that have a more substantial influence on the disparities in mental well-being between groups. These additional factors may include various living conditions, lifestyle habits, or school- or relational related factors.

Studies during and after the period of the Covid-19 pandemic have stressed the negative impact of the pandemic on the mental health of adolescents [46–48]. However, research also highlights a long-term increasing trend in mental health problems among adolescents that started well before the pandemic [33,49,50]. Although few studies have examined trends regarding mental well-being among adolescents, those that have done so, show a decreasing trend [32,33]. Our study confirms this downward trend and does not show any reinforcing trend that can be specifically associated with the period of the pandemic.

There are possible theoretical explanations that could contribute to the understanding of the findings for the non-normative groups. One such theory posits that internalised oppression may underlie feelings and mental schemas of inferiority [51]. This in turn has the potential to exert a negative influence on the mental well-being of individuals belonging to various demographic groups, including girls, ethnic minorities, non-heterosexual individuals, and people with disabilities. The results of this study suggest that explanatory factors may provide a reasonable basis for the observed differences in mental well-being. With that said, it is important to note that these theories are predicated on mental ill-health rather than mental well-being; the relationship between, e.g., internalised oppression and the well-being dimensions of mental health is yet to be investigated.

The study results show significant and increasing differences in mental well-being levels between boys and girls. It has been suggested that a combination of biological, cognitive and contextual factors can explain adolescent girls' higher rate of mental health problems [52]. The same factors could possibly explain the gender difference in mental well-being, although more research is needed on this association. In addition, there is also a risk of measurement and clinician bias, which may result in an under- or overreporting of mental health, thereby potentially clouding the actual differences between gender groups.

A review on gender health offers four explanations for gender differences in mental health: socialisation, different help seeking patterns, different coping strategies and gender stratification regarding resources [53]. These explanations can be understood within theoretical frameworks of gender structure and sex roles. Therefore, the understanding of gender structures among scholars and practitioners is important for the surveillance and reporting of mental health problems and mental well-being. It is also essential for identifying the most effective interventions to address these issues.

A comparison of the SSES groups in the current study reveals a significant association between SSES level and mental well-being. The results indicate that as SSES levels increase, there is a corresponding increase in mental well-being. This outcome aligns with the established association between socioeconomic status or income inequality and ill-health [22,54]. Prior research has shown that inequalities in the prevalence of mental health problems are associated with other

social determinants known from previous research [41]. The current study displays that the same pattern of inequality is applicable to mental well-being as well. Consequently, social groups that already have a challenged health or are at risk of poor health are also deprived of the significant health resource that mental well-being has demonstrated to be [10]. This, in turn, may serve to reinforce other inequalities, as these groups are deprived of a resource that could assist them in mastering everyday life and achieving potential for productivity and personal growth.

Another perspective on the results is the intersection (i.e., combination) of the explanatory factors. Assuming that some groups (e.g., non-normative groups) are more exposed to distress than others and that this leads to lower levels of mental well-being [55], there should reasonably be differences in perceived health between, for example, a heterosexual woman without a disability and a non-heterosexual woman with a disability. In the current study, the aim was not to highlight intersectionality, but rather to isolate the effect of each of the explanatory factors. Nevertheless, there is a need for research on how intersectionality affects mental well-being.

Knowledge of the distribution of mental well-being in the population is therefore important for public health policy and practice. Public health practitioners should take this distribution, and the unequal distribution of resources that follows, into account when designing interventions. To reduce health inequalities linked to the social gradient, interventions should be universal, but with a scope and intensity proportionate to the degree of disadvantage, also known as proportionate universalism [56]. Absence of this consideration in intervention design can lead to the accelerated and widespread adoption of interventions by individuals already possessing greater resources, which in turn risks reinforcing the unequal distribution of mental well-being.

The current study has several strengths. One is that it is based on an extensive material covering a whole population´s well-being over nearly a decade. Because of its size and statistical power, it is possible to detect differences in small groups. These groups are only 2–3% of the total material but still consist of a few hundred individuals. However, the limited size of these groups renders the material sensitive to variations over time, which hampers the interpretations of temporal changes within these groups. On the other hand, this problem is mitigated to a certain extent by employing reverse Helmert's contrast, where comparisons are based on collapsing data from multiple measurements. Another strength is the use of MHC-SF as a psychometrically evaluated instrument that includes both dominant traditions of mental well-being.

However, the study also has weaknesses. One such weakness is the risk of nonresponse bias, as despite a response rate of 75%, which could be considered as relatively high, 25% of the population remained unaccounted for. This potential discrepancy in the sample population could imply that the non-responders and responders may exhibit divergent levels of mental well-being. Additionally, certain self-reported explanatory variables may be regarded as exhibiting potential deficiencies. For example, the self-reported impairment and disorder variables introduce uncertainty about the degree of impairment, as they are not based on clinical assessments.

## Conclusion

In conclusion, this study demonstrates a deterioration in mental well-being over time, predominantly among girls, with a small effect also seen among adolescents with hearing impairment. Girls, non-heterosexual adolescents, and adolescents with low subjective socioeconomic status or impairments exhibit lower levels of mental well-being compared to boys, heterosexual adolescents, and adolescents with higher subjective socioeconomic status or without impairments, respectively. Of these factors, subjective socioeconomic status emerges as the most significant predictor of mental well-being. The collective impact of the studied factors on mental well-being is small. However, from an equality perspective, these differences are concerning, particularly regarding gender, given that half of the population consists of girls. Thus, although the observed disparities are small, their potential implications for public health warrant careful consideration. The results underscore the imperative of promoting mental well-being in adolescents, particularly among vulnerable groups. These results are important for public health practitioners when designing interventions and formulating public health policy.

 

## Supporting information

**S1 Table. Regression coefficients.**
(DOCX)

## Acknowledgments

The authors would like to thank all the students who participated in the study, and Region Västmanland for conducting the data collection. We would also like to thank the internal reviewers and seminar participants at University Health Care Research Centre for valuable input on the manuscript.

## Author contributions

**Conceptualization:** Lena Uvhagen, Johanna Gustafsson, Fredrik Söderqvist.

**Data curation:** Lena Uvhagen, Fredrik Söderqvist.

**Formal analysis:** Lena Uvhagen, Johanna Gustafsson, Fredrik Söderqvist.

**Methodology:** Lena Uvhagen, Johanna Gustafsson, Fredrik Söderqvist.

**Supervision:** Fredrik Söderqvist.

**Visualization:** Lena Uvhagen.

**Writing – original draft:** Lena Uvhagen.

**Writing – review & editing:** Lena Uvhagen, Johanna Gustafsson, Fredrik Söderqvist.

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
