## [Decision Letter · Decision Letter 0]

30 Dec 2024

PONE-D-24-28213Mental well-being in Swedish adolescents 2014-2023: a repeated population-based cross-sectional study focusing on temporal variations and differences between groupsPLOS ONE

Dear Dr. Uvhagen,

Thank you for submitting your manuscript to PLOS ONE. After careful consideration, we feel that it has merit but does not fully meet PLOS ONE’s publication criteria as it currently stands. Therefore, we invite you to submit a revised version of the manuscript that addresses the points raised during the review process.

We look forward to receiving your revised manuscript.

Kind regards,

Angelina Wilson Fadiji, PhD

Academic Editor

PLOS ONE

For additional information about PLOS ONE ethical requirements for human subjects research, please refer to http://journals.plos.org/plosone/s/submission-guidelines#loc-human-subjects-research .

4. In the online submission form, you indicated that [Data cannot be shared publicly because of ethical and legal restrictions concerning the stundent's concent and data ownership. Data can be made available by the authors upon request, for researchers who meet the criteria for access to confidential data.]. All PLOS journals now require all data underlying the findings described in their manuscript to be freely available to other researchers, either 1. In a public repository, 2. Within the manuscript itself, or 3. Uploaded as supplementary information.

Additional Editor Comments:

Dear Lena

Your manuscript has now been reviewed and we will like to invite you to make the following revisions to your manuscript.

This study is interesting as it has a focus on mental well-being in adolescents, considering both differences over time and between groups. Overall, the manuscript is well structured with alignment between different parts of the content. I have, however, some questions and comments mainly regarding the method section.

The title is informative and aligned with aim and research questions. The abstract coherently corresponds to the content in the manuscript. The introduction gives a review of previous research and describes both the knowledge gap and rationale for this study.

In materials and methods, it would be of interest for the reader with justifications for choice of design and choice of statistical analyses (normal distributed data?). A description of the demographics in Västmanland County is of value also for the discussion of generalization of the findings. Further, a description of the reliability and validity of the measures mental well-being and self-reported socioeconomic status is useful to have in the section of material and methods instead of mentioning it in the discussion.

In the result section is missing data described, how was it treated?

Reviewers' comments:

Reviewer's Responses to Questions

**Comments to the Author**

1. Is the manuscript technically sound, and do the data support the conclusions?

Reviewer #1: Yes

2. Has the statistical analysis been performed appropriately and rigorously? 

Reviewer #1: Yes

3. Have the authors made all data underlying the findings in their manuscript fully available?

Reviewer #1: Yes

4. Is the manuscript presented in an intelligible fashion and written in standard English?

Reviewer #1: Yes

5. Review Comments to the Author

Reviewer #1: This study is interesting as it has a focus on mental well-being in adolescents, considering both differences over time and between groups. Overall, the manuscript is well structured with alignment between different parts of the content. I have, however, some questions and comments mainly regarding the method section.

The title is informative and aligned with aim and research questions. The abstract coherently corresponds to the content in the manuscript. The introduction gives a review of previous research and describes both the knowledge gap and rationale for this study.

In materials and methods, it would be of interest for the reader with justifications for choice of design and choice of statistical analyses (normal distributed data?). A description of the demographics in Västmanland County is of value also for the discussion of generalization of the findings. Further, a description of the reliability and validity of the measures mental well-being and self-reported socioeconomic status is useful to have in the section of material and methods instead of mentioning it in the discussion.

In the result section is missing data described, how was it treated?

6. PLOS authors have the option to publish the peer review history of their article (what does this mean? ). If published, this will include your full peer review and any attached files.

**Do you want your identity to be public for this peer review?** For information about this choice, including consent withdrawal, please see our Privacy Policy .

Reviewer #1: **Yes: ** Lene Lindberg

---

## [Author Response · Author response to Decision Letter 1]

7 Mar 2025

Dear Editor and reviewer,

Thank you for reviewing the manuscript Mental well-being in Swedish adolescents 2014-2023: a repeated population-based cross-sectional study focusing on temporal variations and differences between groups, and for considering it for publication in PLOS ONE. Below, we have responded to each point raised by editor and reviewer as follows:

1. The manuscript has been revised to ensure that it meets PLOS ONE´s style requirements.

2. Clarifications regarding the participant consent have been made in the Materials and methods section in the manuscript and in the online submission information. The clarifications mainly concern the information that the participants received before the study.

3a. The Data Availability Statement have been updated and clarified according to point 3a. The clarifications states that data can be shared upon request to corresponding author or a non-author contact person, that public sharing is not possible due to ethical and legal restrictions concerning the student's consent and data ownership, and that the data is owned by a third party who has restrictions.

4. Please see point 3a above.

5. The reference list has been revised to ensure that it is complete and correct. This includes the adding of one reference (#34) as a result of the new paragraph on demographics raised by reviewer, see below.

Additional editor and reviewer comments:

a) Regarding study design, we have clarified the choice of study design to fulfil the overall aim in the first paragraph in the Study design, population and data collection section. The headline has been changed to better match the content of the section.

Regarding choice of statistical analyses, the reasons for choosing Reverse Helmert’s contrast is clarified.

We have also added some information on the assumption underlying the reliability of the performed regression analysis.

In addition to these clarifications, we consider the section Statistical analysis to include a relatively detailed description of each step of the analysis and how it relates to each research questions and hence why it is justified.

b) Demographics of Västmanland county has been added in the Study design, population and data collection section in the manuscript. In the headline of this section, “participants” has been replaced with “population” to include the context.

Additionally, the generalisability of the findings has also been commented on in the Discussion section.

c) Description of validity and reliability of the measures of Mental well-being and Self-reported socioeconomic status has been elaborated and moved from the Discussion to the Material and methods section.

d) The handling of missing data has been described in the Results section.

We hope that these revisions are sufficient and fulfil your expectations. In addition to these revisions raised by editor and reviewer, we also made some copyediting changes after receiving go-ahead from editor via e-mail 24/01/2025. These changes does not include factual changes, only language editing, and have been made mainly in the discussion but also the last paragraph of the abstract, the conclusions and one paragraph in the introduction. To increase readability of the manuscript, these changes are not made with track changes.

---

## [Editor Report · Decision Letter 1]

17 Apr 2025

Mental well-being in Swedish adolescents 2014-2023: a repeated population-based cross-sectional study focusing on temporal variations and differences between groups

PONE-D-24-28213R1

Dear Lena,

We’re pleased to inform you that your manuscript has been judged scientifically suitable for publication and will be formally accepted for publication once it meets all outstanding technical requirements.

Kind regards,

Angelina Wilson Fadiji, PhD

Academic Editor

PLOS ONE
---

## [Editor Report · Acceptance letter]

PONE-D-24-28213R1

PLOS ONE

Dear Dr. Uvhagen,

I'm pleased to inform you that your manuscript has been deemed suitable for publication in PLOS ONE. Congratulations! Your manuscript is now being handed over to our production team.

Kind regards,

on behalf of

Dr. Angelina Wilson Fadiji

Academic Editor

PLOS ONE